# Multicomponent Exercise on Physical Function, Cognition and Hemodynamic Parameters of Community-Dwelling Older Adults: A Quasi-Experimental Study

**DOI:** 10.3390/ijerph16122184

**Published:** 2019-06-20

**Authors:** Ivan de Oliveira Gonçalves, Alexandre Nunes Bandeira, Hélio José Coelho-Júnior, Samuel da Silva Aguiar, Samuel Minucci Camargo, Ricardo Yukio Asano, Miguel Luiz Batista Júnior

**Affiliations:** 1Laboratory of Adipose Tissue Biology, Integrated Group of Biotechnology, University of Mogi das Cruzes, Mogi das Cruzes 08780-911, Brazil; samuelmogi@hotmail.com (S.M.C.); migueljr4@me.com (M.L.B.J.); 2School of Physical Education, College UNISUZ, SP 08675-130, Brazil; allexandrebandeira@hotmail.com; 3School of Physical Education, University of Campinas, Campinas, SP 13083-851, Brazil; helio.j.coelho.junior@gmail.com; 4Rehabilitation unit, Mãe Mariana Nursing Home, Poá, SP 08562-460, Brazil; 5School of Physical Education, Catholic University of Brasília, Brasília 71966-700, Brazil; sdsa10@hotmail.com; 6School of Arts, Sciences and Humanities, University of São Paulo, São Paulo 03828-000, Brazil; ricardoasano1@gmail.com

**Keywords:** exercise, aged, rehabilitation, functional tests

## Abstract

This paper reports on a quasi-experimental study that aimed to identify changes in muscle function (i.e., mobility, maximal walking speed, lower limb muscle strength, balance, and transfer capacity), cognition (i.e., executive function) and hemodynamic parameters of community-dwelling Brazilian older adults during a six-month multicomponent exercise program (MCEP). A total of 436 community-dwelling older adults performed functional, cognitive and hemodynamic assessments before and after a six-month MCEP. The program of exercise was performed twice a week over 26 weeks at moderate intensity. Results indicate that balance, mobility (i.e., usual and maximal walking speeds) and transfer capacity (*p* < 0.05) were significantly improved after the MCEP. Moreover, all hemodynamic parameters (i.e., systolic, diastolic and mean arterial pressures), except for heart rate (*p* > 0.05), were significantly reduced after the intervention. The current findings indicate that a six-month MCEP may provide physical and hemodynamic benefits in community-dwelling older adults. Nevertheless, our findings need to be confirmed in larger samples and better designed studies.

## 1. Introduction

According to the United Nations (UN), the ageing of the world’s population is one of the most important topics in the 21st century, given its implications for nearly all sectors of society, including health, transportation, labor and financial markets, and economy [1]. Nowadays, the older population represents around 900 million people worldwide, and this prevalence is expected to double by 2050, where the number of older people may reach nearly two billion [1]. Nevertheless, it is important to mention that different patterns of the demographic transition of ageing population may be observed according to geographic characteristics, which may suggest that some countries may need more time and attention in the development of public health programs [1]. Projections for Brazil, for example, indicate a 14% increase in the prevalence of people aged 60 years or over until 2050, which is higher than the expected for Northern America (6.6%) and Oceania (6.0%) [1].

The age-related decline in physical and cognitive functions is one of the most remarkable changes observed in older people since these factors are directly associated with the subject’s capacity to interact with the world. Indeed, according to the Pan American Health Organization (PAHO), independence (i.e., physical function) and autonomy (i.e., cognitive capacity) are inherently associated with the concept of health in older adults [2]. Furthermore, the cardiovascular system seems to be extremely assaulted by the aging process, given that blood pressure values increase exponentially according to age [3].

This phenomenon deserves attention because evidence has demonstrated that impairments in physical and cognitive functions are risk factors for the development of functional disability, sarcopenia, frailty, and institutionalization, to mention a few [4,5]. In addition, high blood pressure is a risk factor for the development of cardiovascular and cerebrovascular diseases, as well as early death [6]. Lastly, functional limitations, cognitive impairment, and cardiovascular disease are part of the concept of multimorbid in older adults and, consequently, collaborate with negative health outcomes, which places a considerable economic burden on health care systems [7].

Given the aforementioned observations, the development of therapies to counter-regulate age-related impairments in physical function, cognition and blood pressure has been a priority in public health research in the last decades. Among the numerous proposed therapies (e.g., hormonal replacement and diet), the practice of physical activity has received much attention, given its role as a preventive tool in the context of physical disability, cognitive decline, and development of chronic diseases [8,9]. However, evidence suggests that approximately 50% of the world’s older population are physically inactive (less than 150 min of moderate intensity aerobic activity per week) [10]. Interestingly, a high prevalence of physical inactivity is observed in South American countries, mainly in Brazil, where physical inactivity accounts for more than 60% of the Brazilian older population [11]. These data deserve concern, since physical inactivity was responsible for 6% of the burden of coronary heart disease and 5.3 of the 57 million deaths worldwide in 2008 [10].

Many findings have demonstrated that different regimes of physical exercise—a structured form of physical activity—may improve physical function, cognition, and hemodynamic parameters of older adults. Indeed, older adults have shown improved mobility, transfer capacity, balance, executive function, memory, as well as significant reductions in blood pressure, after resistance and aerobic exercise training [8,9]. Nevertheless, some features of these regimes of exercise training may limit their use in public health programs, such as the high cost to acquire and maintain the equipment, the space needed to distribute the equipment, and the number of health professionals needed to prescribe the exercise [12].

On the other hand, it is possible to suggest that multicomponent exercise programs (MCEP), a kind of exercise training that combines different regimes of exercise (e.g., resistance, balance, gait, aerobic, and flexibility) in the same exercise session [13,14], could be a feasible alternative to public health programs aimed at maintaining or even improve physical and cognitive functions of community-dwelling older adults, since it allows the combination of different exercise regimes in the same exercise routine, thereby not requiring sessions of physical exercise with long duration while developing several physical capacities and skills [13]. Other features of MCEP should be highlighted, such as: (a) its dynamic characteristic, which may collaborate to a high adherence to the exercise program; (b) the sessions of exercise may be performed in groups, collaborating with improvements on social skills; and (c) the program of training can be primarily designed to use low cost equipment (e.g., elastic bands), collaborating with its implementation in any kind of setting (e.g., long-term care) and performance by individuals with or without physical and cognitive limitations.

This hypothesis is supported by a couple bodies of evidence that demonstrate significant improvements in the physical function [14,15,16,17,18], cognition [14,19,20,21], and hemodynamic parameters [22] of older adults who performed MCEPs. However, despite the beneficial effects of the abovementioned studies, most MCEPs are composed only by resistance, balance, and aerobic exercises [2,15,16,19], while few investigations combine many types of exercise [14,18,21]. Other limitations include the small sample sizes; the investigation of older adults with different conditions (e.g., frail) and from different settings; the lack of a pattern regarding the distribution of the exercise regimes in the MCEP, given that some studies prioritized resistance training, while others prioritized gait exercises; and the brief period of intervention. However, with respect to the latter, evidence indicates that minimum periods of intervention (≥6 months) may be necessary to cause cognitive changes in older adults [23,24,25].

Therefore, the present quasi-experimental study aimed to identify changes in muscle function (i.e., mobility, maximal walking speed, lower limb muscle strength, balance, and transfer capacity), cognition (i.e., executive function) and hemodynamic parameters of community-dwelling Brazilian older adults during a six-month MCEP.

Our hypothesis is that changes in the physical functioning, cognitive parameters, and hemodynamic parameters of community-dwelling older adults may be observed after our MCEP.

## 2. Materials and Methods

This was a quasi-experimental study aimed to identify changes in muscle function (i.e., mobility, maximal walking speed, lower limb muscle strength, balance, and transfer capacity), cognition (i.e., executive function) and hemodynamic parameters of community-dwelling Brazilian older adults during a six-month MCEP. All volunteers signed the informed consent form and completed all measurements. This study was approved by the Research Ethics Committee of the University of Mogi das Cruzes and was developed in accordance with the Declaration of Helsinki of the World Medical Association (1964, as revised in 1975, 1983, 1989, 1989, 1996 and 2000) and according to Resolution 196/96 of the National Health Council.

### Study Design

The data presented here are a secondary analysis of data collected as part of a larger quasi-experimental study that investigated the changes in physical performance, cognition, and hemodynamic parameters of community-dwelling older adults with different chronic conditions during a MCEP. The findings regarding hypertensive [12,26], osteoarthritis [27], and diabetic [28] participants were previously published by our group. In these investigations, we found improved physical function in older adults with hypertension, diabetes mellitus type II, and osteoarthritis who practiced our six-month MCEP. In addition, blood pressure was reduced in hypertensive and diabetic people after the MCEP, but not in older adults with osteoarthritis and hypertension. Nevertheless, our MCEP seems to be limited to improving cognition. In the present investigation, the MCEP was investigated in the context of health promotion, regardless morbidities and comorbidities, as a tool to improve or at least preserve autonomy and independence and reduce cardiovascular risk. Therefore, data of published and unpublished studies were polled and analyzed as a single study.

## 3. Subjects

A total of 436 community-dwelling Brazilian older untrained volunteers (♂ = 50 (11.5%)) were recruited from a specialized senior center in a town located in the metropolitan area of São Paulo, Brazil. Volunteers were recruited by convenience and asked verbally by the medical team and researchers about their participation in the study. Candidates were eligible to be part of the present study if they were at least 60-year-old, lived in the community, showed independence to perform the ADL according to Katz Index (6 points), were able to ambulate independently without an assistive device, had no dementia according to age- and schooling-adjusted Mini-Mental State Examination (MMSE) scores [29], and signed the informed consent form. Exclusion criteria were as follows: nursing home residence, changes on pharmacological therapy during the investigation, missing values, physical (e.g., angina) and/or psychological (e.g., fear) discomfort during exercise sessions, pulmonary disease, neurological or psychiatric disease (e.g., Parkinson’s or Alzheimer’s disease), musculoskeletal disorders, any kind of dizziness, blurred vision or light-headedness when rise or remain standing for long, which could indicate orthostatic hypotension and/or labyrinthitis, and were absent from more than three sessions of physical exercise. We also excluded participants who were prescribed hormone replacement therapy and/or psychotropic drugs. The presence of hypertension (HTN), type II diabetes mellitus (T2DM), osteoarthritis, cardiovascular diseases (CVD) and osteoporosis was not considered an exclusion criterion provided that the clinical symptoms were pharmacologically controlled. Candidates were interviewed and examined by a physician and a certified nurse for eligibility criteria.

## 4. Evaluations

All participants were instructed to refrain from any exhausting physical activity for a period of 96-h before and drinking alcoholic and caffeinated beverages 24-h before testing. Although alimentary ingestion was not controlled, participants were instructed to maintain their food intake during the study period. Baseline evaluations were performed 5 days before the beginning of the MCEP. Likewise, the final evaluations were performed on the fifth day after the last exercise session. We followed the methods of Coelho-Júnior et al. [30].

### 4.1. Morphological Measurements

A weight scale with a Filizola^®^ (São Paulo, Brazil) stadiometer was used to measure body mass (kg) and height (cm). The body mass index (BMI) was determined by using the formula body mass (kg)/height (m^2^). An anthropometric tape (flexible and inextensible) (Sanny^®^, São Paulo, Brazil) was used to obtain all measurements (i.e., waist circumference (WC), hip circumference (HC), and neck circumference (NC)). Participants remained in a standing position, head held erect, eyes forward, with the arms relaxed at the side of the body, feet kept together, wearing light clothes. The WC was assessed at the mid-point between the last floating rib and the highest point of the iliac crest. HC was evaluated at the highest point of the buttocks. NC was measured right above the cricoid cartilage and perpendicular to the long axis of the neck.

### 4.2. Functional Parameters

All physical and functional tests were administered by two experienced assessors. One researcher was responsible for detailing the operational procedures, demonstrating the test before the evaluation, quantifying the performance, and evaluating the motor gesture. The other assessor ensured the safety of the participant. After explanation and before tests, volunteers performed a familiarization trial to ensure they understood the test. Then, the volunteers performed all tests twice, and the best result obtained in each test was used in the analysis. The tests were distributed in a room as stations and were performed in a circuited fashion one after the other. A one-minute interval between trials was provided. During all tests, verbal encouragement was provided to ensure that volunteers achieved the best possible performance without compromising safety.

### 4.3. One-leg Stand Test

The one-leg stand test was performed with the volunteers standing in a unipodal stance with the dominant lower limb, the contralateral knee remaining flexed at 90°, the arms folded across the chest, and the head straight [27]. A stopwatch (1/100 second accuracy) was activated when the volunteer raised their foot off the floor and was stopped when the foot touched the floor again. The maximum performance time was up to 30 s, considered the best test result.

### 4.4. Sit-to-stand Test

Volunteers were requested to rise from a chair (total height: 87 cm; seat height: 45 cm; width: 33 cm) five times as quick as possible with arms folded across the chest. The stopwatch was activated when the volunteer raised their buttocks off the chair and was stopped when the volunteer seated back at the end of the fifth stand [27].

### 4.5. Walking Speed Test

Walking speed (WS) was measured over 3 m. This distance was chosen because of space limitations. However, high concordance has been observed between the results recorded on 3 m and 6 m courses [27]. For the test, volunteers were required to walk five meters (including one-meter acceleration and one-meter deceleration) at their usual and fastest possible pace (without running). Before the evaluation, both feet of each volunteer were to remain on the starting line. The measurement was started when a foot reached the 1-meter line and was stopped when a foot reached the 4 m line. The 1 m intervals at the beginning and at the end of the course were used to avoid early acceleration and/or deceleration [27].

### 4.6. TUG Test

The TUG test involves getting up from a chair (total height: 87 cm; seat height: 45 cm; width: 33 cm), walking 3 m around a marker placed on the floor, coming back to the same position, and sitting back on the chair [27]. The subjects started the test wore their regular footwear, with their back against the chair, arms resting on the chair’s arms, and with the feet in contact with the ground. A researcher instructed the volunteers to on the word “go”, get up and walk as fast as possible without compromising safety in the boundary of three meters on the ground, turn, return to the chair, and sit down again. The stopwatch was activated when the volunteer got up from the chair and was stopped when the participant’s back touches the backrest of the chair [31]. A longer time taken to perform the test indicates a lower performance.

### 4.7. Executive Function (EF)—TUG Cognitive Test

TUG cognitive test was accomplished to evaluate Executive Function (EF) [27]. This test is performed with the conventional TUG equally; however, a cognitive task (verbal fluency, animal category) must be accomplished during the motor task. Therefore, after the signal of the evaluator, the volunteer performed the route—stand up from the chair, walk three meters, turn around, walk three meters back, and sit down again—naming as many animals as he/she could remember. This task was performed aloud, allowing the evaluators to confirm if the volunteers were accomplishing the task. The time spans to complete were recorded for evaluation.

### 4.8. Hemodynamic Measurements

The procedures for measurement of blood pressure were adapted from the VII Joint National Committee on Prevention, Detection, Evaluation, and Treatment of High Blood Pressure [32]. In summary, volunteers remained in a sitting position on a comfortable chair for 15 min in a quiet room. After this period, an appropriate cuff was placed at approximately the midpoint of the upper left arm (heart level). An automatic, noninvasive, and validated arterial blood pressure monitor (Microlife-BP 3BT0A, Microlife, Widnau, Switzerland) was used to measure systolic blood pressure (SBP), diastolic blood pressure (DBP), and heart rate (HR). During blood pressure recording, volunteers remained relaxed in the sitting position, with parallel feet at one shoulder width, both forearm and hands on the table, supinated hands, backs against the chair, without move or talk. The volunteer did not have access to blood pressure values during measurement. The evaluation lasted approximately 80 s and was performed three times with 1 min of rest among the measurements. The mean of measurements of each volunteer was used in the final analysis. Mean arterial pressure (MAP) was evaluated according to the following equation: MAP = (SBP + (2 × DBP))/3. The size of the arm cuff was selected after measuring the arm circumference of each participant (Sanny, São Paulo, Brazil). All volunteers were evaluated within the first month after the update of the medical records.

## 5. Multicomponent Exercise Program (MCEP)

The present MCEP was designed according to the definition of Tarazona-Santabalbina et al. [14], which states that a MCEP should include endurance, strength, coordination, balance, and flexibility exercises.

Each exercise session lasted ~60 minutes, and was performed twice a week, on nonconsecutive days, during 26 weeks under the supervision of an exercise physiologist at the fitness center of an institutional center for elderly care and living (Centro de Convivência do Idoso (CCI)), Poá, Brazil. The exercise routine was structured based on the combination of thirteen exercise stations each one composed by a resistance, balance/proprioception, coordination, flexibility exercise plus a brief gait (i.e., aerobic exercise). Six stations were composed by resistance exercises (i.e., five for lower limbs and one for upper limbs), four were composed by balance/proprioception exercises, two were composed by coordination exercises, and one was composed by flexibility exercises. Each exercise station was performed for one-minute, while aerobic exercises were performed for two-minute. The representation of the resistance, balance/proprioception, and gait exercises used in the present study may be observed in the Figure 1. Approximately 50 participants composed each session of exercise. Exercise intensity was controlled according to the rating of perceived exertion (RPE) method using the adapted Borg scale [33] (i.e., CR-10), which was used to ensure that volunteers performed the exercises in the aimed intensity. This scale was composed of eleven numbers (i.e., 0–10) and eight descriptors (i.e., rest; very, very easy; easy; moderate; somewhat hard; hard; very hard; and maximal), which represents the perception of effort of the subject in front of an exercise load. The higher the reported number, the greater the sensation of effort. During the performance of functional —except for balance exercises—and resistance exercises, volunteers were instructed to maintain the physical activity intensity in 3–5—which represents moderate (i.e., 3), somewhat hard (i.e., 4), and hard (i.e., 5) descriptors. To that, a large picture of RPE scale (i.e., 4 m high and 1.30 m wide) was positioned on the wall in the gym’s room. The increase in the exercise intensity was based on alterations in the cadence of the performance (i.e., faster), for functional exercises and walk. Moreover, for resistance exercises, volunteers could use elastic bands (EXTEX Sports, São Paulo, Brazil) and dumbbells to reach the intensity prescribed.

## 6. Statistical Analyses

Normality of data was tested using the *Kolmogorov*–*Smirnov* test. A Student-t test for dependent samples was used to intragroup comparisons. Cohen’s effect size *d* was calculated to assess the magnitude of the results. The effect size was classified according to Rhea [34] for untrained volunteers. The level of significance was 5% (*p* < 0.05) and all procedures were performed using the GraphPad Prism 6.0 (San Diego, CA, USA). Data are shown as mean ± standard deviation (SD), except for Tables 2, which are shown as Cohen’s *d*.

## 7. Results

No adverse events occurred during the sessions of exercise or during evaluations. The subjects were not absent for more than three sessions of physical exercise. The adherence to the physical exercise program was 100% (0 dropouts). Participants characteristics are shown in Table 1. The study sample included 436 (male = 50 participants (11.5%)) community-dwelling older adults. Volunteers presented an overweight/obesity phenotype according to BMI classification [35]. WC, HP, and NC values indicated a high cardiovascular risk in the studied sample [36]. The morphological parameters were not changed in response to the MCEP. Regarding the hemodynamic parameters, our findings suggest that SBP and DBP values were within a prehypertensive classification. These results are probably the product of an equilibrium between the proportion of hypertensive (58.9%) and normotensive (41.1%) participants. The most prevalent disease after HTN was osteoarthritis (31.7%), accompanied by osteoporosis (25.2%), T2DM (17.9%) and CVD (9.2%).

Physical and executive functions are shown in Figure 2. Significant changes in balance (one-leg stand (+46.5%)), mobility (usual (+16.7%) and maximal (+55.6%) walking speeds), and transfer capacity (TUG (−2.9%)) were observed after the MCEP. However, muscle strength (sit-to-stand) and executive function (TUGcog) were not altered. ES evaluation indicated a small classification for one-leg stand, and usual walking speed; accompanied by a trivial classification for sit-to-stand, maximal walking speed, TUG, and TUG-cog (Table 2).

Hemodynamic parameters are shown in Figure 3. SBP (−4.42%), DBP (−6.68%), and MAP (−5.17%) were significantly changed after the MCEP. HR (+ 0.39%) did not show significant changes. ES evaluation was classified as trivial, regardless of *p* value (Table 2).

## 8. Discussion

The main findings of the present study indicate that significant changes in balance, mobility, transfer capacity, and blood pressure may be observed after a six-month MCEP composed by strength, balance/proprioception, coordination, and aerobic exercises.

Indeed, balance (one-leg stand (+46.5%)), mobility (usual (+16.7%) and maximal (+55.6%) walking speeds), and transfer capacity (TUG (−2.9%)) were significantly improved after our MCEP. These findings are in concordance with earlier studies that investigated several kinds of exercise training, including resistance [37,38] and power training [37]. Regarding MCEP, findings from our group [26,30] and other groups [14,15,16,17,18] have demonstrated that different designs of MCEP may improve physical function of older adults from different settings (e.g., community and long-term care facilities) with various comorbidities, such as frailty, HTN, T2DM, osteoarthritis, osteoporosis, and myocardial infarction.

In an elegant investigation, Freiberger et al. [16] found increased transfer capacity, balance, mobility, and lower-limb muscle strength in community-dwelling older adults who practiced a 24-month MCEP composed by aerobic, balance, and resistance exercises. These data are supported by Toto et al. [18], Kang et al. [17], and Ansai et al. [15], who also observed significant improvements in the physical function of community-dwelling older adults submitted to MCEPs.

It is worth mentioning that these studies varied widely regarding the primary variables, including the sample size, MCEP design, time of intervention, number and type of outcomes measures, etc., thus limiting comparisons across studies. Nevertheless, it should be stressed that, except for the trial of Freiberger et al. [16], most investigations have been performed in low sample sizes [14,18,19,20,21,22] and using short time of intervention (e.g., four weeks).

On the other hand, our study had a longer time of intervention (i.e., six months) and investigated a large sample (i.e., 436 volunteers) of older adults. In addition, our MCEP may be further explored in randomized clinical trials (RCT) as an interesting and profitable design of MCEP, given that some changes observed in the present study have not been found in other investigations. Indeed, in the trial of Ansai et al. [15], TUG performance was not changed after the 16-month MCEP. However, we observed that some beneficial changes in physical function may occur earlier than the reported by other studies. Freiberger et al. [16], for example, found that improvements in mobility, lower body muscle strength, and transfer capacity occurred at six months, balance at 12 months, and mobility at 24 months. On the other hand, significant changes in balance, transfer capacity, and mobility were observed after six months in the current study.

A possible explanation for these findings might be that, although some studies have proposed a MCEP, most were only composed by resistance, balance, and aerobic exercises [15,16,19,22], and the last two kind of exercises were poorly prioritized. In the present study, although resistance exercise was prioritized, we proposed a MCEP with a high volume of aerobic, balance, coordination and flexibility exercises. Therefore, combining different kinds of exercise training, as proposed by Tarazona-Santabalbina et al. [14], seems to be essential for the effectiveness of the MCEP [14].

The findings of the present study may have high external validity if confirmed by RCT studies, given that interesting changes in physical performance tests associated with a poor prognosis in older adults [39,40] were observed after our public health group-based MCEP.

The beneficial effects of the current MCEP may not be restricted to physical function since SBP (−4.42%), DBP (−6.68%), and MAP (−5.17%) were significantly changed after the MCEP, suggesting a reduced cardiovascular risk in these participants, given that slight decreases in SBP and DBP reduce cardiovascular and cerebrovascular risk [41]. These findings are supported by prior reports that exercise training, including MCEP, may collaborate in the management of blood pressure in patients at high and low cardiovascular risk [22,42].

Interestingly, a larger magnitude of changes in SBP and DBP was observed in the present study when compared to a recent meta-analysis [42] regarding the effects of endurance training (SBP = −3.5 mmHg and DBP = −2.5 mmHg), resistance training (SBP = −1.8 mmHg and DBP = −3.2 mmHg), and combined training (SBP = −3.5 mmHg and DBP = −2.2 mmHg). This phenomenon has been observed in other investigations in previous studies as well [22], given that a substantial decrease in SBP (6 mmHg) and DBP (2 mmHg) has been observed after a three-month MCEP.

Taken together, these findings indicate that multicomponent exercise should be widely explored in better designed studies, since it can be in the near future one more public health alternative in the management of blood pressure in older adults. Indeed, in addition to its effectiveness, MCEP seems to be easier to apply, cheaper, and more feasible when compared to other regimes of exercise, such as resistance and aerobic training.

Despite the significant changes observed in physical function and hemodynamic parameters, executive function was not altered after our MCEP. These results add evidence regarding the effects of MCEP on the cognitive function of older adults, given the previous studies reported conflicting results. In fact, many studies have demonstrated improved cognitive function in older adults after MCEP [14,19,20,21]. These data are supported by a recent systematic review [20], which demonstrated improved memory, attention, verbal fluency, inhibition, and global cognitive scores after MCEPs. The lack of consistency among the studies is probably due to the differences in the composition of the samples and MCEP designs. The findings of de Asteasu et al. [20] were based on cognitively healthy older adults and older adults with cognitive impairment, while the investigations of Suzuki et al. [19] and Tarazona et al. [14] were based on older adults with mild cognitive impairment and frailty, respectively. Regarding community-dwelling older adults, similar to the present study, Ansai et al. [15] investigated cognitively healthy older adults and did not observe significant changes in TUG-cog performance. Nevertheless, the authors found a worse performance in these participants when they were compared to volunteers who performed resistance training or remained sedentary for three months. Therefore, these data may suggest that our MCEP conferred a protective effect against executive dysfunction. However, the lack of a control group limits our interpretation.

Other methodological limitations should be mentioned to collaborate with future studies, such as a more detailed cognitive and scholar profile of the volunteers, the low prevalence of men, the inclusion of other regimes of MCEPs, and samples composed by volunteers from different settings (e.g., long-term care facilities) and conditions (e.g., sarcopenic), since we investigated community-dwelling older adults able to perform the ADL and ambulate independently without assistive device.

## 9. Conclusions

The current findings indicate that a six-month MCEP may provide physical and hemodynamic benefits in community-dwelling older adults. These findings indicate that multicomponent exercise may be a public health alternative in the management of mobility impairment and blood pressure in older adults, since, in addition to its effectiveness, MCEP seems to be easier to apply, cheaper, and more feasible when compared to other regimes of exercise, such as resistance and aerobic training. Nevertheless, our findings need to be confirmed in larger samples and better designed studies.

## Figures and Tables

**Figure 1 ijerph-16-02184-f001:**
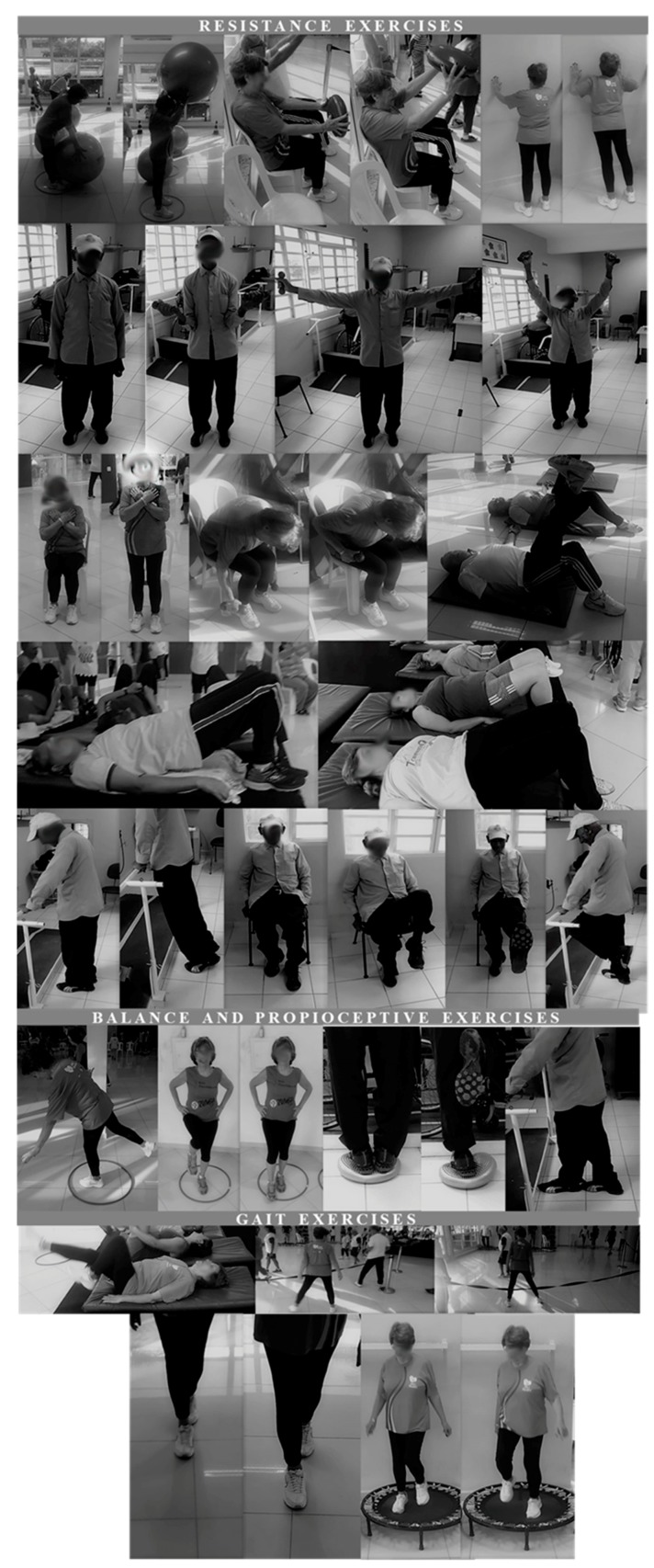
Representation of the resistance, balance/proprioception, and gait exercises used in the present MCEP.

**Figure 2 ijerph-16-02184-f002:**
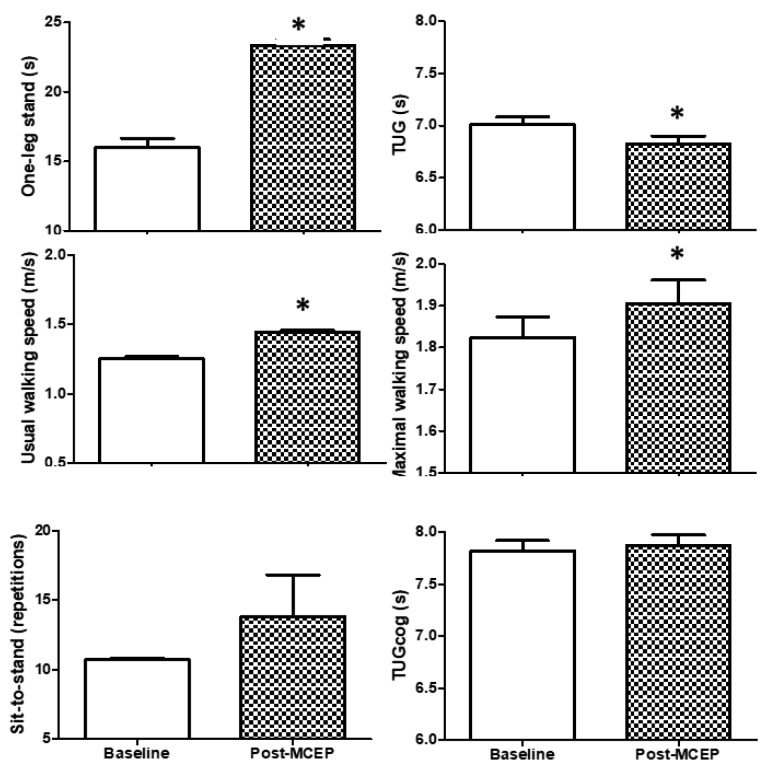
Physical and executive functions. Values are mean ± S.D. TUG, Timed “Up and Go”; TUGcog, TUG with a cognitive task; * *p* < 0.05 versus baseline.

**Figure 3 ijerph-16-02184-f003:**
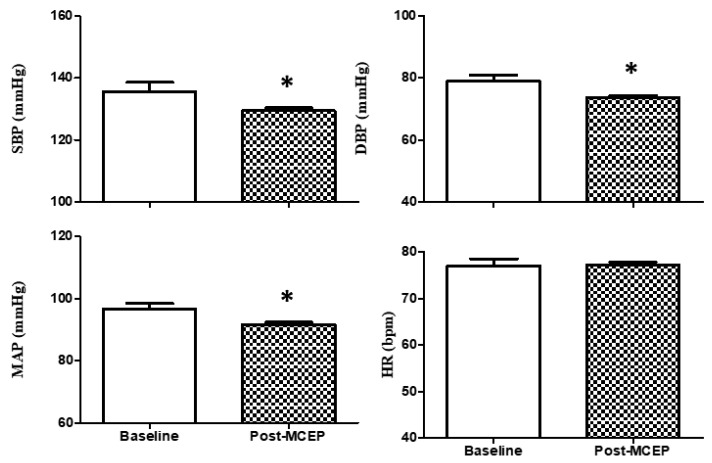
Hemodynamic parameters. Values are mean ± S.D. SBP, systolic blood pressure; DBP, diastolic blood pressure; MAP, mean arterial pressure; HR, heart rate; * *p* < 0.05 versus baseline.

**Table 1 ijerph-16-02184-t001:** Overall characteristics of the studies sample.

Variables	Older Adults (*n* = 436; ♂ = 11.5%)
Age (years)	66.0 ± 5.9
Morphological	
Weight (kg)	69.9 ± 14.0
Height (m)	1.56 ± 0.17
BMI (kg/m^2^)	28.7 ± 7.3
WC (cm)	97.5 ± 12.2
HC (cm)	106.3 ± 46.9
NC (cm)	36.8 ± 3.6
Physical function and cognition	
One-leg stand (s)	15.9 ± 12.5
Sit-to-stand (repetitions)	10.7 ± 13.6
Usual walking speed (m/s)	1.25 ± 0.29
Maximal walking speed (m/s)	1.82 ± 1.01
TUG (s)	7.0 ± 1.5
TUG-cog (s)	7.8 ± 2.0
Hemodynamic	
SBP (mmHg)	135.6 ± 63.7
DBP (mmHg)	79.0 ± 39.1
MAP (mmHg)	96.7 ± 35.9
HR (bpm)	76.9 ± 32.5
Morbidities (%)	
Hypertension	58.9
Osteoarthritis	31.7
Osteoporosis	25.2
Diabetes mellitus type II	17.9
Myocardial infarction	9.2

Values are mean ± S.D. BMI, Body mass index; WC, Waist circumference; HC, Hip circumference; NC, Neck circumference; SBP, Systolic blood pressure; DBP, Diastolic blood pressure; MAP, Mean arterial pressure; HR, Heart rate; TUG, Timed Up and Go; Cog, Cognitive.

**Table 2 ijerph-16-02184-t002:** ES of functional, cognitive, and hemodynamic parameters.

Variables	Older Adults (*n* = 436)	
Functional and cognitive		
One-leg stand	ES	−0.65 (small)
Sit-to-stand	ES	−0.67 (small)
Usual walking speed	ES	−0.66 (small)
Maximal walking speed	ES	−0.07 (trivial)
TUG	ES	0.12 (trivial)
TUG-cog	ES	−0.02 (trivial)
Hemodynamic		
SBP (mmHg)	ES	0.13 (trivial)
DBP (mmHg)	ES	0.17 (trivial)
MAP (mmHg)	ES	0.18 (trivial)
HR (bpm)	ES	−0.01 (trivial)

Cog, Cognitive; DBP, Diastolic blood pressure; ES, Effect size; HR, Heart rate; MAP, Mean arterial pressure; SBP, Systolic blood pressure; TUG, Timed Up and Go.

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
