# Peer review of "Multicomponent Exercise on Physical Function, Cognition and Hemodynamic Parameters of Community-Dwelling Older Adults: A Quasi-Experimental Study"

_ijerph, 2019, doi:10.3390/ijerph16122184_

Round 1

Reviewer 1 Report

Ni further comments

Author Response

April 2019

Dear Editor

We are sincerely grateful for this new opportunity to improve and submit our manuscript to review. We are also grateful for the Reviewers for their precious time. The changes required by the Reviewer 3 may be observed highlighted in yellow in the main manuscript. 

I hope to hear from you soon.

Ivan de Oliveira Gonçalves, EP, MSc

Reviewers 1, 2, and 3.

(x) English language and style are fine/minor spell check required

Answer: Dear Reviewers, please observe that English language was revised one more time.

Reviewer 3

I think that the authors have done a good job in revising the manuscript and it has markedly improved. However, I feel that the use of the term 'effects' in the title and no mention of the study design in the abstract or introduction puts the reader on the wrong foot. I would suggest that the authors either mention their observational design earlier on in the paper and avoid terms as 'effects' or 'changes/improvements induced by' and replace them by e.g. 'changes occurring during the intervention' to avoid any misunderstanding of the causal associations that can be inferred from the current study design.

Answer: Dear Reviewer, we totally agree with you. Please observe that many changes were performed in the manuscript in attempt to perform the changes that you suggested. Noticeably, we replaced all indications that changes/improvements/reductions were induced by our MCEP by changes were observed after our MCEP. In addition, we mentioned that this was a Quasi-experimental study in the title and abstract sections.

Reviewer 2 Report

The paper has been improved and I appreciate the time the authors took to resolve the issues and consider the suggestions that have been made. I am glad they resubmitted the paper! The paper reads well!

Author Response

(The authors gave the same response as above.)

Reviewer 3 Report

I think that the authors have done a good job in revising the manuscript and it has markedly improved. However, I feel that the use of the term 'effects' in the title and no mention of the study design in the abstract or introduction puts the reader on the wrong foot. I would suggest that the authors either mention their observational design earlier on in the paper and avoid terms as 'effects' or 'changes/improvements induced by' and replace them by e.g. 'changes occurring during the intervention' to avoid any misunderstanding of the causal associations that can be inferred from the current study design.

Author Response

April 2019

Dear Editor

We are sincerely grateful for this new opportunity to improve and submit our manuscript to review. We are also grateful for the Reviewers for their precious time. The changes required by the Reviewer 3 may be observed highlighted in yellow in the main manuscript. 

I hope to hear from you soon.

Ivan de Oliveira Gonçalves, EP, MSc

Reviewers 1, 2, and 3.

(x) English language and style are fine/minor spell check required

Answer: Dear Reviewers, please observe that English language was revised one more time.

Reviewer 3

I think that the authors have done a good job in revising the manuscript and it has markedly improved. However, I feel that the use of the term 'effects' in the title and no mention of the study design in the abstract or introduction puts the reader on the wrong foot. I would suggest that the authors either mention their observational design earlier on in the paper and avoid terms as 'effects' or 'changes/improvements induced by' and replace them by e.g. 'changes occurring during the intervention' to avoid any misunderstanding of the causal associations that can be inferred from the current study design.

Answer: Dear Reviewer, we totally agree with you. Please observe that many changes were performed in the manuscript in attempt to perform the changes that you suggested. Noticeably, we replaced all indications that changes/improvements/reductions were induced by our MCEP by changes were observed after our MCEP. In addition, we mentioned that this was a Quasi-experimental study in the title and abstract sections.

This manuscript is a resubmission of an earlier submission. The following is a list of the peer review reports and author responses from that submission.

Round 1

Reviewer 1 Report

This is a manuscript presenting what I believe are secondary/ancillary outcomes from a large one-armed study of MCEP in a sizable population of older adults attending group classes in the community. 

I think a few changes to the manuscript would be helpful for the reader.

First, because this is an analysis of what I believe are secondary outcomes from this study I think it's important to both be explicit about what these outcomes represent (primary or secondary) and provide some background on what was found in the prior reports (I noted in methods that you have previously reported on hypertensive, diabetic and OA outcomes.  Presumably you identified a single primary outcome for a large trial like this.

It would be helpful to specify how you assessed "significant cognitive impairment". 

I'm sure that the prior papers show this but I think it's essential that you provide additional information on the MCEP. Can you design a schematic to show the exercises, their rotation, and the progression? I struggled to understand what you meant by MCEP. The way it is described it sounds more like functional training, rather than exercise. This is fine but it's not what I would call a traditional exercise program. Ansai and Rebellato for example, sounds like it is a more traditional aerobic and strength regimen. I wouldn't call this MCEP as you implement it in this study. It seem, rather, that you are defining MCEP as including functional training and aerobic walking.

I don't believe the use of Cohen's d is appropriate in this instance. My understanding of Cohen's d is that it is for comparison of two groups. I believe you need an effect size estimate that accounts repeated measures. 

I also think it would be helpful to provide change in your measures over time in Table 1.  Perhaps you could combine Table 1, 2, and 3, and include follow-up testing. 

Reviewer 2 Report

This is a good study with well written results and easy to understand tables.  I only have several points that should be addressed. 

The introduction is well written and provides a solid base for the study. However, I expected to read some information relevant to the country of Brazil. Aging is diverse. However, the effects of aging sometimes are not.  If the authors are going to make reference to what is going on worldwide, they should also speak to the similarities/differences of the worldwide numbers to that of Brazil. This will add value to understanding the importance of physical activity among all older persons.

Additionally, to make the paper more appealing and novel, the fact that the intervention was 6 months (and the importance of this- as many people who start an exercise program quit within 6 months) should be emphasized. 

Lines 59-62 are good information but the authors do not have a reference for the one sentence (that should not be a paragraph, alone).  I suggest either adding more information, with references. Alternatively, it could be placed with the following paragraph on MCEP- but authors still need some information with references to support that sentence. Additionally, they could use lines 340-343, in the conclusion, to bridge the thought process.  

Table 1, Overall Characteristics (starting line 265) should have the number of men/women in the study (as there is mention of 11.5% being men on line 104 on page 2). 

Within the Conclusion, another limitation is the number of men in the study. It is possible that this is common for these types of studies. However, it is difficult to know because many authors choose referenced (e.g. Ansai)  not to disclose the gender of their participants. 

Reviewer 3 Report

This is a paper reporting changes in older people participating in a functional training program conducted in the community. The study has a large sample size, but studying the effects of the intervention (as the authors want) would require a different study design. A randomized controlled trial showing changes over time in the intervention and a control group would be needed to determine intervention effects. The current study could, however, test the feasibility of the intervention in the community-setting, which would also be a valuable contribution to the current research literature. Unfortunately, the authors have left out large parts of the information needed for this study to be a feasibility study and many of their conclusions are not justified because of their flawed design and focus on determining intervention effects. As such I consider this paper not acceptable for publication and encourage the authors to change their perspective in their presentation to a feasibility study (see e.g. SPIRIT (https://www.spirit-statement.org/) or CONSORT (http://www.consort-statement.org/) for reporting guidelines on intervention studies). Below my suggestions for improving the quality of the study in order to make it suitable for publication.

Major comments:

- In the introduction as well as in the discussion it would be helpful if the authors would provide more details on training regimen used in earlier studies and how this compares to their study. It is not very useful to state that the methods are different (e.g. resistance training, balance training, aerobic exercise, functional training, etc.), but please point out what is the difference. Also the basis of the exercises conducted in the current study is unclear. The term multicomponent is not very meaningful unless at least some details of its different components are mentioned. In addition, to me it is not clear whether the term multicomponent refers to different types of functional training (which is to what the authors frequently refer to in other parts of the paper) or also to other components.

- The list of exclusion criteria also contains criteria related to the uptake of the intervention. If the authors would decide on making this a feasibility study (as I recommended) exact numbers of participants having missing values, discomfort during the training sessions, and poor compliance to the program (>3 sessions missed) would be needed. These numbers can not be used as exclusion criteria, but are valuable numbers describing the feasibility of the implementation of this program in the community-setting.

- Related to the previous point, in the results, the authors stated that participants were not absent for more than 3 sessions, which is the case by definition due to the exclusion criteria they used. The reader needs to know how many participants actually started the program, but were absent for more sessions.

- This paper states that volunteers are recruited for this study from a specialized healthcare centre (p.2, l.105). Yet the authors state in the discussion that their sample is representative of the Brazilian population (p.8, l.316). This does not seem to be justified based on the information provided in the paper. Or do the authors have any other evidence not presented in the paper, e.g. comparison of the composition of their sample with general population?

- Please add a flow chart of how many people were asked to participate in the study and numbers of eligible participants, numbers of people unwilling to participate and numbers of people excluded for different reasons. This information is crucial also for determining the representativeness of the sample.

- More details are needed on the exact program conducted. The authors for example state that the results are “likely highly reproducible and have high external validity”, yet due to lack of detail, it is impossible to repeat this study. E.g. the authors describe that the exercises were changed during the program, do they mean that there was a progressive increase in its intensity or load? Or were the movements changed? Please specify. Also it would be helpful to understand whether the exercises were aerobic or more resistance type of exercises (author describe use of resistance bands and loads, but no specifics).

- The discussion requires a complete revision, taming down the conclusions based on study effects, which can not be assessed with the current study design, and discussing more relevant aspects of the intervention and its value in a community-based setting.

Minor comments:

- In the introduction the authors refer to a hypothesis (p.1,l.76), but to me it is not clear what this hypothesis is. 

- How was the cognitive impairment used as inclusion criterion (p.2,l.110) determined?

- The amount of details varies greatly between the different measures of functioning. E.g. the characteristics of the chair used are mentioned for the TUG test, but not for the repeated sit-to-stand test. Similarly the accuracy of the stopwatch is mentioned only once, but not in the first test describing use of this device.

-  Please check the description of the walking test. It says that the length of the walkway was 3m (p.3,l.165), but also 5m (p.3,l.168). Furthermore, it is unclear whether participants started on the start line (l.169) or whether there was room for acceleration (l.171).

-  To aid the reader with results interpretation it would be helpful to organize the results section into separate sections describing 1) participant characteristics, 2) feasibility of the program, e.g. details on the compliance to the program, 3) changes in outcome measures (irrespective of whether a change was seen or not).

- This paper would hugely benefit from language editing.